# Public School Choice Options in the United States

**Shelby L. Smith \***  **, Margaret Dawson-Amoah, Tong Tong**  **, Nicolas Pardo, Elizabeth Ann Alonso-Morris and Adam Kho \***

Rossier School of Education, University of Southern California, Los Angeles, CA 90089, USA; dawsonam@usc.edu (M.D.-A.); ttong059@usc.edu (T.T.); pardonic@usc.edu (N.P.); morrisel@usc.edu (E.A.A.-M.)
\* Correspondence: shelbysm@usc.edu (S.L.S.); akho@usc.edu (A.K.)

**Definition:** Under the structure of compulsory education, students in the United States are required to attend school until at least 16 years of age, which can be done at a variety of educational institutions, both public and private. Amongst public schools, students are each assigned a neighborhood school but also frequently have the option to attend a choice school. While the purpose of neighborhood schools is to provide a guaranteed educational option that accommodates most students, choice schools serve varied purposes that accommodate specific learning styles and societal goals. Four types of publicly funded choice schools are magnet, charter, alternative, and virtual schools. While each was established to serve a specific societal goal, their purposes have shifted over time and have produced varied student outcomes, both academic and non-academic.

**Keywords:** school choice; choice schools; magnet schools; charter schools; alternative schools; virtual schools

## 1. Introduction

Compulsory education in the United States can be completed at a variety of educational institutions. Given that children are required to attend school until at least 16 years of age [1], the government provides a tuition-free option in the form of traditional public schools (TPS), which students are assigned to based on residence. However, given that quality and programs vary widely between schools [2], not all families prefer to enroll their children in their assigned TPS and may choose to enroll in a different school. Though not all families participate in school choice for a myriad of reasons (e.g., lack of transportation, awareness of options), school choice in the United States aims, in theory, to provide families with the opportunity to choose the school that best suits their student's educational needs, thus funding their attendance with public money, as is the case with TPS [3].

In order to attend a different school, some families opt to move to a neighborhood with a different assigned TPS. In this case, families leverage residential mobility to gain access to their preferred school that is publicly funded. Movement for this purpose is known by many names, including neighborhood choice, unofficial choice [4], and Tiebout sorting [2]. However, residential mobility is not accessible to all families due to financial constraints and the strong, positive correlation between home prices and school quality [5]. Based on this relationship, neighborhood choice is a practice that is not equally available to all families and that is utilized most often by higher-income families [5].

Based on inequitable access to neighborhood school assignments and the existent diversity in school quality, students are not required to enroll in their assigned TPS. Rather, school choice provides families with other options that allow for public funding to be used at both publicly and privately operated schools [3]. The expanded opportunity for parents to exercise choice in education is argued to promote equity by not only loosening the association between housing choices and school attendance [5] but also incentivizing higher educational quality [6] and re-centralizing the priorities of all parents and students in education [7]. These benefits of school choice are meant to operate through families' selection of their preferred school, thus creating competition amongst surrounding schools [6].

The schools that families have an opportunity to select from are known as choice schools. Amongst the types of choice schools available, there are options that are publicly financed and governed besides TPS, such as magnet, charter, alternative, and virtual schools. These options are not only intended to promote educational equity and quality but also serve a variety of social and economic purposes. While the United States also has non-public choice schools, such as private schools and homeschooling, these schooling options differ in that they predominantly serve private interests rather than societal ones. Public choice schools specifically serve as "agencies of the society as a whole" (p. 32) [7] because the institutions are accountable to the government based on their public funding and governance. However, given that societal needs continuously change, the goals of choice schools have also evolved since they were established. The following sections provide an overview of the societal purposes that public choice schools were intended to serve, how these missions have changed over time, and how the missions and administration of choice schools have driven differential outcomes for students.

## 2. History of School Choice

Prior to the establishment of choice schools in the mid-1900s, school choice solely functioned to facilitate movement between TPS and private schools. Following the adoption of compulsory schooling in the mid-1800s [8], families had the option to enroll their student in TPS, pay for private school, or exercise neighborhood choice to gain entry into their preferred TPS. However, as previously discussed, neighborhood choice was only accessible to families with financial means for residential mobility. Given the limitations of neighborhood choice, select states began to adopt voucher programs in the mid-1800s to extend choice and allow families to use public funds for private school attendance [9,10].

While choice policies emerged in the early stages of public education, choice was not immediately accessible to all groups in the United States. The segregated nature of education was also embedded in choice policies; thus, only White families had the opportunity to engage in choice to improve student-to-school matches. For example, some of the first voucher policies were solely established in predominantly White states, such as Maine and Vermont [9,10]. Based on the selective adoption of choice policies in these states, vouchers were only functionally available to low-income White children in rural areas [9,10]. At the same time, racially minoritized and dis/abled students experienced intersectional discrimination and exclusion, which was later referred to as "educational debt" because of its compounding and long-term negative effects [11]. Educational debt was accumulated not only through exclusion from choice policies, but also relegation to under-resourced schools and systems that further marginalized students' identities [12–15].

Exclusion and de jure segregation on the basis race in the United States began to be legally challenged in 1896 and was ruled unconstitutional by the Supreme Court in 1954 with the case of *Brown v. Board of Education* [16]. However, segregation in all public spheres, including education, was continuously carried out through discriminatory practices and policies. In practice, housing was segregated due to red-lining and racially restrictive covenants that excluded non-White families from home-buying in certain neighborhoods [17]. Based on these discriminatory housing practices, racially minoritized families were further excluded from the practice of neighborhood school choice. Additionally, while vouchers could in theory be utilized to circumvent neighborhood patterns of segregation, in practice they were advocated for and utilized as tools to maintain segregated schools in Southern states during the post-Brown era [18,19].

Despite the continuing contribution of some choice policies to segregation, school choice has also been conceptualized as a way to promote integration and diversity. Federal policies and priorities, such as the Civil Rights Act of 1964, the Americans with Disabilities Act of 1990, the Individuals with Disabilities Education Act of 1990, and the accountability era of the late 1990s and early 2000s, were seminal in the reconceptualization of school choice as a tool to broaden access among historically marginalized populations to quality schools [20]. Furthermore, Milton Friedman's 1955 essay "The Role of Government in

Education" proliferated the argument that school choice facilitates greater competition among schools and creates a fairer and more meritocratic education system [6]. These developments and arguments have since contributed to the creation of choice schools, each sector of which are intended to achieve various societal goals. The following sections outline the intended goals of magnet, charter, alternative, and virtual schools and how these missions have shifted over time.

## 3. Magnet Schools

In 1968, magnet schools were established as the first public choice school system as a response to the Civil Rights Movement. To prompt desegregation, specialized academic programs were established in segregated schools to attract voluntary enrollment of White students into primarily Black schools [21–24]. This model was first utilized in Tacoma, Washington and was found to successfully shift school demographics from 90% African American to just under 50% within two years [25,26].

In tandem with the goal of desegregation, magnet schools were also established to promote educational innovation through unique programming and school structures. The specialized programs at magnet schools can have a variety of foci, such as science, technology, engineering, and mathematics (STEM), performing and/or visual arts, or health and medicine [27]. In addition to introducing specialized curricula, magnet schools also diversify schooling options through their possible structures. While magnet schools can be established as stand-alone schools, which are known as full magnets, they can also be integrated as programs within existing schools and are known as partial magnets [28,29]. In both cases, enrollment can either be open to all students or be restricted based on criteria, such as tests or portfolios [30]. This structural variety not only contributes to diversified options for students, but also flexibility for districts that seek to establish magnet schools.

To encourage the establishment of magnet schools in districts across the United States, the federal government established the Magnet Schools Assistance Program (MSAP) in 1985. Under the mandate of MSAP, Local Education Agencies are eligible for funding to establish and operate magnet schools as a desegregation effort [31]. In addition to MSAP, various federal education policies have allocated funds for magnet schools, including USD 100,000,000 under No Child Left Behind (NCLB) in 2001 and USD 108,000,000 under the Every Students Succeeds Act (ESSA) in 2015. These funding efforts highlight continued support for magnet school expansion across the United States.

The use of magnet schools to encourage desegregation is an effort that can either be pursued voluntarily by school districts or under court mandates [32,33]. In preemptive measures, some districts will voluntarily establish magnet schools to curtail White flight and encourage, rather than force, integration [34]. However, when a court does find that a district is in violation of laws that prohibit segregation, they may require that magnet schools be established [32,33]. Often, court decisions to require the establishment of magnet schools are driven by the perception that they not only contribute to desegregation but also educational innovation [35].

### 3.1. Current Landscape

While magnet schools pursue both desegregation and academic goals, the advent of new policies concerning race have created limitations to the pursuit of desegregation. In 2007, the ruling in the case of *The Parents Involved in Community Schools v. Seattle School District No.1* prohibited the use of race in school assignments at the K-12 level [36]. This ruling created ambiguity around how magnet schools should seek the reduction of racial isolation without racially driven admissions decisions [36].

Based on the new precedent of color-evasiveness in federal policy and judicial rulings [37,38], the purpose of magnet schools has shifted away from voluntary desegregation, and the focus has been siloed on academic innovation. While the ruling in 2007 necessitated that schools end race-based admission practices, the focus of magnet schools had already begun to shift away from desegregation. In 1996, it was found that only 42% of magnet

schools that benefited from MSAP funding had clear desegregation goals [39]. In 2003, a study also concluded that some magnet schools experienced resegregation, as indicated by racial isolation [40]. These trends indicate that magnet schools are responsive not only to official policies but also societal pressures that led to the gradual de-prioritization of desegregation goals at some schools.

However, the degree to which the goal of desegregation has been deprioritized or upheld varies across the magnet school sector. Across the literature, there have been accounts of both increased and decreased integration at magnet schools [41–43]. Decreased integration can occur through either over-enrollment of racially minoritized or White students. Over-enrollment of White students is specifically prevalent at magnet schools with criteria-based enrollment, also known as selective magnets [21,30]. The overrepresentation of White students in these programs has been attributed to a variety of factors, including selection bias in the motivation of applicants and practices of cream skimming [44]. In addition to issues of racial stratification, selective magnets also have, on average, the lowest enrollment of students who benefit from free or reduced lunch [27]. Despite the issues of stratification associated with selective magnets, they constitute a minority of magnet schools across the United States [27].

Overall, the magnet school sector has experienced rapid growth in the last two decades. Specifically, Statistica reports that the number of magnet schools increased from 1469 in 2001 to 3497 in 2020 [45]. As of the 2021–2022 school year, magnet schools enroll over two million students and account for approximately 4.5% of public school enrollment in the United States [46].

### 3.2. Student Academic Outcomes

Though they serve a large number of students in public schools, magnet schools represent a relatively underdeveloped area of research. In 1983, the United States Department of Education conducted a nationwide study that described academic gains for magnet school students [47]. Specifically, the study found that 80% of magnet schools produced higher reading scores than their TPS counterparts and 41% produced higher math scores [47]. Despite this clear picture of achievement, more contemporary studies have found both positive [42,48–50], negative, and null effects [46,51–54] for magnet school students. In several studies, positive relationships have been found between magnet school attendance and academic achievement [42,48–50]. However, in one study of math and reading achievement, the relationships became nullified or negative when student characteristics and prior achievement were controlled for [46], suggesting selection effects. In other studies, magnet school attendance has also been predominantly associated with null or negative effects [51,52], except for at partial magnets, where positive effects have been observed in both reading and math [51].

While the study of students' academic scores in certain subjects, such as math and reading, produce uncertainty regarding the efficacy of magnet schools, studies point to the production of other consistently positive outcomes. For example, one study found positive associations between magnet school attendance and the outcomes of advanced course-taking and graduation rates at the high school level [50]. Additionally, Kemple and Snipe [53] found in a nationwide study between 1993 and 1999 that magnet school students deemed high-risk accrued more credits toward graduation than their TPS counterparts. When examined at a school level, magnet schools have also been found to be positively correlated with overall achievement [50]. However, when analyses have been disaggregated by race, findings suggest that Students of Color may not experience the same beneficial improvements as White students [55]. This finding is important considering that societal pressures have shifted the focus of magnet schools almost solely toward academic achievement despite the initial purpose of desegregation and improved schooling for racially minoritized students.

## 4. Charter Schools

While magnet schools have faced societal pressures to shift purposes, the market-oriented goals of charter schools have remained consistent over time, which has supported the rapid expansion of the sector. The first state law governing charter schools was passed with bipartisan support in 1991 [56]. This law and other state laws to follow have permitted the provision of publicly funded education under private operational structures. In contrast to schools overseen by local education agencies, as with TPS, charter schools are operated by private groups of individuals who enter into contracts with state-approved authorizers. Based on these contracts, charter schools become responsible for student achievement, and the school can be closed by the authorizer if outcomes are unsatisfactory [57].

Under this structure, charter schools have embodied free market principles since their inception that are intended to promote innovation and improve education. The incorporation of market principles into education was first promoted by the previously discussed economist Milton Friedman, who argued that reducing the role of government and increasing individual choice in school selection would increase marketplace competition between schools [58]. Such competition is intended to theoretically induce innovation at charter schools and encourage improvement at TPS to avoid losing students to charter schools [58]. Innovation is specifically garnered through charter schools' operational autonomy from the government, which allows the schools to be responsive to local needs [59] and act swiftly in the implementation of changes [60]. This inducement of innovation and competition has been argued to lead to better schooling options, especially for low-income families and Students of Color, who have been continuously siloed to underachieving urban schools [61].

Charter schools have diversified the schooling options available to students based on both their operational structures and foci. The private operation of charter schools is not monolithic; rather, it can be carried out under a Charter Management Organization (CMO) or an Education Management Organization (EMO), or it can be a free-standing/independent school [57,62]. While CMOs and EMOs differ in their profit structures, both allow for replication of charter school models across several locations in comparison to the often unique nature of free-standing charter schools [63]. Across each of these operational structures, there is also some diversity in the foci that charter schools take. For example, categories used to describe the array of schools include no excuses charters [64], prestige charter schools [65], mission-oriented charters, and market-oriented charters [66]. These categories are not exhaustive but highlight the potential for diversity that charter schools are intended to introduce into the education market.

### 4.1. Current Landscape

Since 1991, charter school legislation has been passed in 45 of 50 states [67], and charter schools have become the most well-known and controversial choice school option in the United States. The expansion of charter schools has been promoted by a number of federal policies, including No Child Left Behind (NCLB) and the Race to the Top (RTTT) competition. Specifically, in 2001, NCLB intentionally acted as a vehicle for school privatization through the portrayal of public schools as failing based on high-stakes testing [68,69]. Based on the claims of a failing TPS system, charter schools were promoted as an alternative that would spur innovation and achievement [70]. In 2009, grants available under RTTT incentivized states to increase or remove caps on the number of charter schools allowed [71]. For instance, in response to receiving an RTTT grant, Tennessee increased their number of charter schools from 29 to 98 between 2010 and 2015 [72]. While not all states received RTTT grants, the policy's campaign was still effective in increasing widespread commitment to charter school expansion [73].

The commitment to charter schools and their expansion has also been reinforced by ideals of high-stakes accountability. While high-stakes accountability is conceptualized separately from marketization, the two movements frequently act to reinforce one another. The adoption of high-stakes accountability efforts in the United States that call for improved

outcomes have acted as an impetus for charter school expansion, which promises to provide innovative solutions to achievement gaps [74].

While charter schools are intended to improve academic outcomes through innovation, the operational structure utilized by some schools has contributed to the marginalization of communities [74–78]. In contrast to TPS, which must make changes through public bureaucratic systems that incorporate community input, charter schools are privately operated by people who often may not be representative of the community or students served [76,77]. This disconnect between the community served and those who operate charter schools is scrutinized for diminishing the liberatory nature of community input in education [74–78].

Additionally, the efficacy of charter schools in diversifying schooling options has also been widely scrutinized. Studies that have aimed to examine if charter schools are truly hubs of innovation have found mixed results, with some affirming the innovative nature of charters [79] and others finding that they use traditional practices and approaches and that their operation is not distinguishable from TPS [59,80]. Notably, one study found that while there has been an increase in "specialist" charter school mission statements over time, they constitute a diminishing proportion of charters [81]. This suggests that while some charter schools act as hubs of innovation, a growing number operate similarly to TPS.

As the charter sector continues to grow, as of 2021, there were 2045 CMO-managed charter schools, 680 EMO-managed charter schools, and 5088 free-standing charter schools in the United States [82]. Across all kinds of charter schools, Black and Hispanic students are overrepresented compared to their TPS and magnet school counterparts [27,63,83]. In 2015, 41% of charter schools had enrollments of over 80% non-White students, twice that of TPS [84]. This trend contributes to the evidence that the expansion of charter schools has changed the educational market for historically marginalized students over the past few decades by increasing racial stratification, especially for Black students [83,85]. Overall, as of the 2021–2022 school year, charter school students account for approximately 7.5% of public school enrollment in the United States [86].

## 4.2. Student Academic Outcomes

The literature that seeks to evaluate the impact of charter schools for these students has predominantly compared reading and math performance of charter school students with their own performance in traditional public schools in prior years or other students' performance in traditional public schools. Studies that examined school-level outcomes of charter schools generally agree that charter schools tend to struggle in the first year but improve over time to perform on par or even better than traditional public schools [87–90]. However, studies that evaluate charter schools based on student-level performance are more varied, with reported positive [88,89,91,92], negative [87,93,94], and null effects [95–97]. It is worth pointing out that studies that have found positive effects tend to be lottery-based studies [98], which require charter schools to be over-enrolled to be included, meaning family and student interest in these schools exceed their enrollment capacity. Over-enrollment in charter schools can be indicative of their academic success, and thus the findings might not be generalizable across all charter schools. Besides academic outcomes, other research has also demonstrated that charter schools positively impact students' postsecondary outcomes [87,90], as well as behavioral and health outcomes [92,99,100].

Within the charter school sector, however, there is variation in performance. A recent report by Stanford's Center for Research on Education Outcomes (CREDO) has found that CMO-affiliated charter schools outperform their stand-alone counterparts across reading and math when compared to the growth observed in local TPS [101]. This finding suggests that student growth in math and reading is more greatly supported at CMO-affiliated schools than at stand-alone charters. In addition to school structure, findings regarding the impacts of charter schools are also dependent on student demographics. For instance, research has found that effective charter schools improve the performance of minoritized student groups, including racially minoritized and socioeconomically dis-

advantaged students [91,102,103]. One study found that in Massachusetts, urban charter schools improved math and English language arts outcomes for Students of Color and low-income students [91]. However, evaluations of no excuses charter schools, which are often defined by the enforcement of strict disciplinary codes and extended school days and years, have shown that these schools have a detrimental impact on the experiences of Students of Color, lead to exclusionary practices, and inhibit the implementation of restorative justice practices [60,104,105]. Collectively, these findings suggest that students' experiences and outcomes at charter schools vary greatly.

## 5. Alternative Schools

Amongst the array of choice schools that diversify schooling options for students, alternative schools were intentionally established to serve a variety of students under different missions. Alternative schools were established in the 1970s to provide options to students whose needs were not met in TPS [106,107]. Defining alternative schools can be challenging due to their diversity in goals, structure, target populations, and services offered. Given the diversity of the sector, various classification systems have been adopted; one example defines four types of alternative schools: typical, vocational, special education, and juvenile detention [108,109]. While special education and juvenile detention schools are distinguished by the students they serve, vocational schools are distinguished based on their specialized offerings of technical or career education. In addition to this classification system, another describes three types of alternative schools: innovative choice schools, "last stop" behavior or academic focus schools, and remedial or emotional focus [110]. Across the two classification systems, the categories are neither exhaustive nor mutually exclusive; however, they highlight that alternative schools are intended to serve students who have been classified as both over- and under-performing [111], as well as those with goals that deviate from the options available at TPS [106,107].

While alternative schools provide programs and learning environments to benefit a diversity of students, not all students are enrolled by choice. Generally, if a student wishes to enroll in an alternative school, families can express a desire to enroll, but the school staff must agree or criteria must be met (e.g., teen parenthood, discipline) based on the mission of the school [112]. Alternatively, other students are mandated to attend a given alternative school on the basis of disciplinary problems, disability accommodation, refugee status, prior academic performance, or English language proficiency [112–115]. In one study, 65% of state education administrators indicated that student placement at an alternative school is predominantly decided by someone other than the student or their family [112].

Students who do attend an alternative school are often provided with options not otherwise accessible at TPS. For example, in lieu of or in addition to a high school diploma, some alternative schools provide options for GED attainment or pathways to transition back to TPS [116–118]. Additionally, instruction at alternative schools is typified by individualized attention, smaller classes, flexibility, and other supportive features [117] that are not inherently incorporated into all TPS. The features and options are intended to carry out the mission of alternative schools, which is to serve students whose needs are not met in TPS [106,107].

### 5.1. Current Landscape

Despite the broad and diverse missions adopted by the earliest alternative schools, the sector's focus began to narrow in the 1980s to focus on the provision of remedial education and enrollment of low-performing students [119]. This shifted focus has been reified by a rise in state policies that explicitly define alternative education as school for "at-risk" students [114]. Across all types of alternative schools, it has been found that enrollment is largest in typical alternative schools, which frequently have behavioral or therapeutic missions [108]. In addition to the growth in enrollment in typical alternative schools, the sector has also been promoted for the education of students with dis/abilities, who are often stigmatized. This shifted focus was partially prompted by the Individuals with

Disabilities Education Act (IDEA) of 1990, which allowed alternative schools to be used as interim settings for students with dis/abilities under specific conditions [120].

While alternative schools with different missions vary in their racial composition, there is a general overrepresentation of Students of Color compared to enrollment at TPS [109]. As of the 2021–2022 school year, alternative schools account for a little over 1% of public school students [1]. Of students at alternative schools, 77% are high schoolers, and an approximately equal portion of the remainder attend middle and elementary schools [108]. Across the schools that have both disciplinary and academic foci, there is an underrepresentation of White students and overrepresentation of students who are Black, Hispanic, and who have dis/abilities [109]. Specifically, Black students are overrepresented by 23 and 14 percentage points at disciplinary and academic alternative schools, respectively; in these gaps, overrepresentation is driven by the enrollment of Black males [109].

The overrepresentation of racially minoritized students drives serious issues for equity in alternative schools. Due to the focus of alternative schools on "at-risk" or low-achieving students, the schools and their students have widely been stigmatized. In a study of middle school boys, McNulty and Roseboro [121] found that attending alternative schools reinforced labels of the students being "bad kids." Furthermore, it has been found that larger educational communities and TPS students perceive alternative school students as troublemakers or as those who do not place adequate effort in school [122,123]. This stigma further marginalizes Students of Color and those with dis/abilities because they are typically overrepresented at alternative schools.

*5.2. Student Outcomes*

For students who do attend alternative schools, there is scarce evidence regarding academic outcomes. Assessment of changes in student achievement is complicated by both the diversity in the sector and the high mobility of alternative school students. For students who do transition back to TPS, one study in California found that after a year, approximately 60% were still at their TPS, 17% returned to an alternative school, and 23% were no longer enrolled in either an alternative school or TPS [124]. For students who maintain enrollment at alternative schools, studies have found both positive and negative effects on students' academic achievement [125,126] and null effects on long-term outcomes [122]. However, it should be noted that the Texas-based study that found negative effects on achievement compared alternative school students to those at TPS [125]. While "at-risk" indicators were incorporated into the analysis, students at TPS may still systematically differ from those at alternative schools and be inadequate comparisons. Despite the ambiguity in evaluating achievement metrics, a study that surveyed alternative school graduates did find that 94% of students felt that enrollment improved their academic performance [127]. This finding suggests that despite a lack of evidence on academic achievement, alternative school students generally have positive perceptions and experiences within the sector.

Overall, research has widely validated that students' perceptions of their alternative schools are generally positive despite the stigmatization of their schools [127–131]. Students' positive perceptions have been found to be driven by small class sizes, supportive staff, individualized attention, cultures of respect, increased responsibility, and care received from teachers at alternative schools [127–131]. However, positive student perceptions are not consistent across all types of alternative schools. Specifically, some disciplinary alternative schools have been critiqued on the basis that their codes of conduct mirror carceral practices [132,133]. Such practices harness environments of high control [132,133] that are contradictory to the increased responsibility that students desire and value at other alternative schools [127,129,130]. The divergent experiences of alternative school students highlight that while the sector has shifted to focus on students who are seen as "at-risk" [119], it remains diverse in its offerings.

Similarly to the diversity of environments created within the sector, alternative schools also produce various non-academic outcomes based on their goals. Within the sector at

large, it has been found that behavioral interventions have a moderately positive effect on students' behavioral outcomes [134]. However, at alternative schools with a disciplinary focus where enrollment is mandated, a meta-analysis found that disciplinary sanctions only decrease during enrollment, and the schools produce no long-term effects on students' behavior [135]. Again, these findings highlight that the experiences of alternative school students are not monolithic but rather school-dependent.

## 6. Virtual Schools

While the sectors of choice schools reviewed thus far all conform to the tradition of brick-and-mortar schools, virtual schools are a relatively new type of choice school that offer remote learning options to students across the United States. The first virtual schools in the U.S. were Virtual High School (VHS) and Florida Virtual School (FLVS), which were established in 1997 and provided both supplemental and full-time online curricula. These schools were created with the vision to provide more affordable and equitable opportunities for high-quality education to rural, underserved, and at-risk student populations [136]. Overall, virtual schools provide safety, accessibility, teacher availability, and academic fit for students who leave traditional schools due to various reasons, such as bullying, academic struggles, lack of adequate support, and legally mandated accommodations [137].

The benefits and flexibility of virtual schools are available to students not only through public options, but also through schools that are operated by CMOs or private organizations [138]. While virtual schools contribute to diverse schooling options through innovative programs [139–141] similar to other choice schools, the diverse options for school management also uniquely contribute to added diversity in the array of schools available in the United States. While private virtual schools fall outside of the scope of public choice schools, charter and public options still encapsulate a diverse array of options. For example, public virtual schools can be administered at the district or state level [139,141,142], and a single district virtual school may serve multiple districts, whereas a TPS is geographically constrained [141]. Virtual charter schools also exhibit flexibility in enrollment and have acted as a driving force in the expansion of virtual schools, accounting for over 58% of the total enrollment in virtual schools in the 2020–2021 academic year [143]. Among those in virtual charter schools, nearly half are enrolled in schools operated by for-profit EMOs [143].

### 6.1. Current Landscape

While the early expansion of virtual schools was partly attributable to the prevailing advancement of communication technologies and accompanying mindset shifts around learning [136,144,145], the expansion has since been accelerated by the COVID-19 pandemic [138]. Based on growing capabilities of using technology and access to technology, students and the larger society have started to believe that learning can be "an anytime-anywhere experience" [136,144,145]. The disconnect between TPS offerings and this mindset has driven an increased demand for virtual schools [144]. This new demand has ultimately led to not only the expansion of virtual schools but also a shift away from the initial purpose of the sector, which was to provide educational services to rural, underserved, and at-risk student populations [136]. Virtual schools now operate to provide a flexible learning platform to all students, regardless of background or location of residence.

Rather than providing education to underserved students, a series of annual reports from the National Education Policy Center (NEPC) has consistently found that virtual schools enroll a lower percentage of racially minoritized, low-income, special education, and English language learner students compared to the national average [138,144,146–149]. While the reasons for these enrollment disparities are unclear, the literature not specific to virtual schools highlights more limited access to internet in these communities and rural settings more generally [149–153]. The most striking disparity in virtual schools is in the enrollment of special education students, which was half of the national average in the 2019–2020 school year [138] despite reports of students switching from traditional schools

to virtual schools due to issues with mandated accommodations [154,155]. This disparity, along with others, highlights that the mission of virtual schools has not only shifted away from its original purpose but that it has also shifted to enrollment of predominantly White students [138,156].

Overall, enrollment in virtual schools was growing steadily prior to the coronavirus pandemic, and this growth has been further accelerated since 2020. Parents enrolled their children in virtual schools with the belief that these more experienced schools would know how to execute online learning better [157]. Parents have stated that virtual schools have been beneficial, citing flexibility, independence, and safety as the main reasons [158], with virtual options being particularly well-fit for children who have health-related issues [159]. Consequently, these changes were directly reflected in the enrollment numbers of virtual schools. During the 2019–2020 school year, there were a total of 477 full-time K-12 virtual schools enrolling over 330,000 students in the U.S., and an additional 150,000 students were enrolled in blended schools, which are schools with a combination of online and in-class instruction [138]. Relative to the 2017–2018 school year, blended and virtual school enrollment in the U.S. has grown by over 50,000 students. As of the 2021–2022 academic year, virtual schools now account for a little over 1% of public school enrollment in the United States.

### 6.2. Student Outcomes

Research on the effects of full-time virtual schools on student outcomes is limited and remains mostly descriptive, but what exists shows negative results [138,160]. In the most recent national virtual school report from NEPC, Molnar et al. [138] showed that of 338 virtual schools with available performance ratings, only 42.8% were rated as acceptable in the 2019–2020 school year, a 5.7 percentage points drop from the 2017–2018 school year. However, the report did suggest that data around virtual school performance are limited by disparate reporting and accountability requirements [138]. When the available data were disaggregated by models of virtual schools, charter and for-profit-operated schools received lower percentages of acceptable performance ratings than other types of virtual schools, at 35.2% and 37.2%, respectively [138]. Overall, the research has consistently shown that virtual schools perform worse than traditional schools, and the effects on student sub-groups are null to negative [161–166]. Specifically, research has found negative [161,162] and null [163] effects for racially minoritized students and students eligible for free or reduced-price lunch and negative results for English language learners and special education students [161,164–166]. These findings suggest that while virtual schools have increased access to curricula online, access to quality schooling that promotes student success has not necessarily been improved via virtual schools.

### 7. Discussion and Conclusions

Overall, choice schools are both a response and contribution to diversity between schools. Across the United States, quality and programs vary widely between schools [2], and historical systems of exclusion have further exasperated differential access to quality TPS options [10,12–15,17]. Inequitable access to educational resources for Students of Color and those with dis/abilities has led to the accumulation of educational debt [11], which choice schools were intended to partially remedy.

In summary, the four sectors of public choice schools in the United States were established with the following goals:

- Magnet schools were established to encourage desegregation by attracting the voluntary enrollment of White students in schools that predominantly serve Students of Color [22–24].
- Charter schools, as privately operated institutions, were theoretically intended to prompt innovation and generate competition in the education market, thus improving all schools [6,62].

- Alternative schools were founded to serve students whose needs, either academic or social, were not met in TPS [106,107].
- Virtual schools were created to provide more affordable and equitable opportunities for high-quality education for rural, underserved, and at-risk student populations [136].

As a whole, the advent and expansion of these choice school sectors were intended to level the playing field for marginalized students across the United States. However, these school sectors are not immutable, which has led to both the intentional and unintentional minimization of this equity-oriented goal over time.

Since their creation, choice schools have widely expanded the options available to students, both historically marginalized and not. This growth in schooling options is partly attributable to the diversity that is encompassed in each sector of choice schools. While magnet and alternative schools are diverse in relation to the different programs offered [27,108,109], diversity in charter and virtual schools is predominantly characterized by operational structures [57,59,138]. In addition to the vast array of programs and school structures available in these sectors, they are also thought about differently by varying levels of government, which has led to distinct policies by sector. However, while often distinct in their policies and adoption, each category of choice school is not mutually exclusive. For example, a virtual school can also be a charter school, and a magnet program can be part of a larger TPS [28,29,143]. These examples are not exhaustive, but they highlight that the diversity of choice schools spans far beyond what can be encompassed by the titles of magnet, charter, alternative, and virtual schools.

Nevertheless, it should be noted that not all choice schools operate in ways distinguishable from TPS [83], and while the theoretical diversity of options has been intended to promote increased choice for historically marginalized families, societal barriers persist in the post-Brown era that limit the opportunity for choice schools to contribute to equity. As previously noted, societal pressures have shifted the focus of magnet and virtual schools away from their intended purposes of serving historically marginalized students and toward offering unique programs to all students [144], which does not inherently beget equitable access. Additionally, it is seen in alternative schools that when servicing marginalized communities remains as the focal mission, stigma is attached to the given schools [121–123], which further marginalizes students. Based on these changes to the missions of magnet, virtual, and alternative schools, it can be observed that attempts at promoting equitable schooling in the United States create tensions in society that lead to the dilution or appropriation of these efforts. Furthermore, the case of charter school expansion highlights a societal preference for race-neutral and achievement-focused projects.

In addition to constraints on equity and racially minded missions, the possibility for choice schools to contribute to equity is also constrained by inequitable access to the schools themselves. Access to school choice is not inherent but rather a practice that must be actively engaged in by families. Similarly to the exclusion that exists in neighborhood choice based on financial resources [5], enrollment in all choice schools is in practice limited to families who have resources, time, and knowledge to find and enroll their students in the best school match [167,168]. Attendance at choice schools is also constrained for some families when the school does not provide transportation or has requirements for parent involvement [167,168]. Based on these constraints, school choice is most frequently engaged by affluent White families who opt out of schools that predominantly serve racially minoritized students [169]. This practice has led to some racial stratification at choice schools; for instance, virtual schools serve predominantly White students [138,156], whereas racially minoritized students are overrepresented at alternative and charter schools [27,63,83,109]. While some choice schools do note increases in racial integration [41,42], issues of stratification that have been evidenced in other choice schools [83,85] should be heavily investigated due to the academic benefits associated with racially integrated learning spaces [43].

The overall contributions of choice schools to academic achievement and the remedying of educational debt for historically marginalized students vary by sector and have

not been consistently proven across studies. For magnet, charter, and alternative schools, studies have inconsistently suggested that the schools produce positive, null, and negative effects on student achievement. While these findings paint an uncertain portrait of these schools' academic impacts, racially disaggregated analyses suggest that each sector differentially impacts racially minoritized students. Across the literature, it is seen that racially minoritized students do not experience increases in academic achievement at magnet schools [55] but do at charter schools [91,102,103], though this may, in some cases, come at the expense of other outcomes and experiences. Alternatively, virtual schools consistently produce lower high school graduation rates than the national average [138], but they also serve predominantly White and affluent students [138,146–148], which complicates the understanding of equity in choice schools. However, there are other metrics of achievement and support for students that should be considered in the evaluation of equity in choice schools. For example, both alternative and virtual schools have provided educational spaces for students who lacked adequate support or faced social issues in TPS [106,107,137], but some charter schools have utilized practices that are detrimental to the experiences of Students of Color [60]. These effects should be continuously evaluated because, similarly to the provision of rigorous academic programs, the provision of affirming educational spaces also contributes to the remedying of educational debt.

Moving forward, further research is necessary regarding both the academic and non-academic benefits of choice schools for marginalized students to ensure the pursuit of educational equity through choice. While choice schools inherently diversify schooling options, their equity-oriented goals require intentionality to ensure realization. As demonstrated, the equity goals of many choice schools have been diluted over time as the sectors respond to new policies and societal pressures. To ensure equity is continuously pursued, there needs to be political and public attention to how the goals of choice schools are changing over time.

**Author Contributions:** Multiple authors contributed to this article. Individual contributions were provided as follows: Conceptualization, S.L.S. and A.K.; Writing, S.L.S., M.D.-A., T.T., N.P. and E.A.A.-M.; Editing: A.K. and S.L.S.; Supervision, A.K.; Project Administration, S.L.S. All authors have read and agreed to the published version of the manuscript.

**Funding:** This research received no external funding.

**Institutional Review Board Statement:** Not applicable.

**Informed Consent Statement:** Not applicable.

**Data Availability Statement:** Data are contained within article.

**Conflicts of Interest:** The authors declare no conflicts of interest.

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
