# Peer review of "Public School Choice Options in the United States"

_encyclopedia, doi:10.3390/encyclopedia4010006_

Round 1

Reviewer 1 Report

Comments and Suggestions for Authors

This entry about school choice in the U.S. includes a lot of interesting material, but still needs a bit more work.  The author has chosen to focus on four main types of school choice: magnet schools, charter schools, alternative schools, and virtual schools.  Although magnet schools and charter schools have received significant attention in the literature on school choice, the other two have received less attention. 

Note, as an aside, that the very first sentence of the paper is not correct.  Education is not compulsory through secondary school but rather until a student is either 16 or 18, depending on the state.   

The term “choice school” is potentially confusing – and probably shouldn’t be in the title

The author begins by correctly noting  that many families have long been able to choose schools through decisions about where to live or by moving their children to  private schools.  but that those choices are limited to families  who have the financial resources to make them.  The author then clarifies that they are using the term “choice school” in this entry to refer to schooling options that are not related directly to residential decisions but that still are under  “public governance.”  Although they never define the term  “public governance,” I assume it is designed to clarify that they are not writing about private schools or voucher programs.   They go on to claim they are focusing on programs specifically designed to promote public interests and not just private interests.  Given that charter schools are operated by private entities, some of which are for-profit entities, however, I wonder how charter schools fit their definition of choice schools.   Further, I wonder why choice schools, as they define them, do not include the many other schooling options often available within districts such as vocational or technical schools.

The structure of each section is clear and the author does a good job of highlighting mission changes over time but some of the references are outdated

For each category of choice school, the author provides some history, describes the current landscape, and then summarizes impacts on school achievement.  That structure works well.

The discussion of magnet schools is generally solid, but some of the references are outdated. See ,for example, the references on page 5 Line 205 to outcomes in magnet schools vs. traditional public schools. Those references were published in 2000 and 2003 and could well be out of date. Fortunately ,the following paragraphs – and also the prior one --cites more up do date research

In the charter schools discussion, the author should refer to EMOS as well as CMOS.  More importantly,  I question the author’s conclusion at the end of the section about the efficacy of charter schools in serving their purpose to improve academic outcomes for all students.  What about the CREDO studies and the other many recent research studies?  More care needed to highlight the fact that some charter schools are very good but others are very weak, and many end up being shut down.  Also what about how charter schools disrupt the making of good education policy more generally?

The discussion of alternative schools is an interesting addition to standard discussions of schools of choice, but as the authors correctly note, not all students are enrolled by choice.  Again some of the references to studies of student outcomes are dated  – 5 of them are before 2005.

The discussion of virtual schools is generally fine, except that some of the references are outdated.  I find it a bit strange that many of those listed under current landscape are from 2005.  Presumably a lot has happened since then, including during the pandemic. Should there be  more references to the pandemic period? Also I would like to see some data on the percentage breakdown between public and private virtual schools.

Concluding section.

The reference in the final paragraph to how choice school increases market efficiency is a bit misleading and needs to be qualified.  The evidence on efficiency is unclear at best since the availability of choice schools, especially charter schools, interferes with the efficient operation of the overall education system.

The writing throughout is wordy, sometime is not clear,  and needs a thorough editing.

I encourage the author to remove wordy phrases such as “ in an effort that can either be”

And to avoid the passive voice, such as  “ … has been argued to lead ..… “ or .. have been found to be positively… “              Rather than schools  overseen by local education agencies…”

Definitions needed in several places – E.g  “educational debt”  or STEAM

A revised version of the introductory section would clarify and improve the  entry.

Comments on the Quality of English Language

The entry needs to be carefully edited.   See my general comments.

Author Response

Thank you, Reviewer 1, for your comments. Please see attached document for our responses.

Reviewer 2 Report

Comments and Suggestions for Authors

The chapter/entry reads more like school choice cheerleading than scholarly literature.

While the authors carve out private and homeschooling options, the entry/chapter should be retitled to clarify that the content is focused on charter schools rather than choice, broadly.

In the intro, the authors operate from the assumption that school choice is a widely accessible option available to any family who simply does not like their TPS.  This fails to problematize transportation issues (most private and charters do not provide transportation), volunteer hour requirements, religious litmus tests, etc.  The authors dismiss residential relocation as a form of school choice on the grounds of costs (which is a reasonable thing to critique) but fail to apply the same financial constraints to the private sector.

In section 2, the authors explore how school choice has exacerbated segregation (citing DeAngelis and Erickson) – I wonder why the authors did not include a citation from DeAngelis where he adamantly supports segregation via school choice?

Discussion of Friedman’s vision for schools without government is accurate but it fails to mention that Friedman explicitly pushed for vouchers/choice as a way to establish “segregation academies” in the post-Brown era as a way for White families to avoid integration.

In the charter section, the authors put forth the idealized vision of charter schools but ignore (either willfully or unintentionally) the absolute mountain of peer-reviewed literature accumulated over the past 30 years that illustrate the ways in which charters exacerbate racial and class segregation, how they do not outperform TPS on achievement or attainment metrics, how they engage in attrition without replacement through counseling out students, etc.  Again, rather than reading like a balanced overview of the sector, the chapter simply hits some of the highlights from the privatized pro-charter movement.  There is mention of Thernstrom and Thernstrom’s no-excuses approach but zero discussion on the negative or racist impacts of this approach (see Horn, et al). 

The virtual school section fails to mention internet connection whatsoever and takes access – especially in rural areas – for granted.

In the conclusion section, the use of Lubienski, 2004 along Friedman as the quintessential citation seemingly giving credibility to the claim of the benefit of charters for improving educational marketplaces is curious – considering the mountain of work by Lubienski that has problematized charters at each turn.

The phrase “the case of charter school expansion highlights societal preference for race-neutral and achievement-focused projects” ignores, again, the mountain of research concluding charter locations (using GIS methodology – actually by Lubienski and others), that charters massively exacerbate racial segregation (Frankenberg and others), that pro-charter groups like those cited here (EdChoice) as well as other right-wing groups such as AEI have admitted that choice exacerbates segregation but saying it was okay as long as that is “what the educational marketplace wanted.”

In sum, the piece reads as if it were written 20 years ago when much of this was unknown and discussion of choice – namely charters – operated from bumper-sticker slogans and idealized theories.  The citations presented in this writing appear to be cherry-picked to suit an ideological purpose of presenting choice as a beneficial practice while ignoring the bulk of the research that has found, over and over again, that not only are academic achievements worse (a few are mentioned but presented as equal to those that find some positive gains) or that the original intent of choice was racial segregation and that choice has actually lived up to that desire espoused by Friedman.

Author Response

Thank you, Reviewer 2, for your comments. Please see attached document for our responses.

Reviewer 3 Report

Comments and Suggestions for Authors

I enjoyed reading this manuscript and believe it will be a useful reference for interested readers. It should be published, although I do have a few suggestions I would like to see implemented first. 

1. I noticed several citations that were incomplete, either with "n.d." in the place of a year or just "cite". 

2. The opening paragraph says that "school choice thus allows families to choose the school that best suits their student's educational needs." While that may be one of the goals of school choice, it's very theoretical. How many students in choice schools are actually in the school that "best" suits their needs? I don't think it's many. I recommend editing to explain that such is the dream of school choice, but not necessarily the reality (as you demonstrate later). 

3. I understand that vouchers are beyond the scope of the paper, but there's a very fuzzy line between charters and private schools, isn't there? Both are privately run, and when vouchers are implemented (as they are more and more these days), the main difference is that private schools aren't held accountable by government. It might help to add a line in the introduction that defines what your scope is. 

4. When you introduce "educational debt" please explain what it is. The concept is important for people to understand the political debate over choice. 

5. It might be good to explain more clearly what level of government tends to use what tool. Districts use magnets but usually not charters. States use charters but usually not magnets. 

6. This is up to the editor, but usually we see the names of court cases in italics. See section 3.1. 

7. I find the statement "the ruling in 2007 solidified the unlawful nature of race-based admissions" a bit biased. Yes, it's true that the practice was declared unconstitutional. But this wording seems harsh and reveals the authors' ideological perspective. 

8. In that same section on page 4 further down (lines 179 to 185, the phrasing is very awkward, especially the second one. 

9. I'm confused by the line "the mission of charter schools has remained consistent over time which has allowed for the rapid expansion of the sector." First, charter schools have a wide variety of missions. Second, I think they have multiple missions. Third, is their growth really a result of consistency, or of other factors? I think the line should be reconsidered. 

10. Similarly, the authors claim that "the mission of virtual schools has ...shifted away from it's original purpose". I thought the purpose was and continues to be to offer an alternative to brick and mortar schools. What's changed? 

11. I'm bothered by the line on page 11 that says, "the advent and expansion of these choice school sectors was intended to level the playing field for marginalized students across the United States." I think this claim ignores numerous other purposes of school choice, including, arguably, the exact opposite. 

12. The next paragraph (lines 536-542) overstates the situation (and uses the word "vast" too many times). There is research suggesting that school choice, charters in particular, has led to very little actual innovation. While schools might adopt a label, they tend to do business as usual. I think the authors should temper their claim a bit, and perhaps even cite such research.  

Comments on the Quality of English Language

I think the quality of English is excellent, but as I mention above there are some sections of awkward phrasing and exaggerations. 

Author Response

Thank you, Reviewer 3, for your comments. Please see attached document for our responses.

Round 2

Reviewer 2 Report

Comments and Suggestions for Authors

Edits resolve previous concerns.

Reviewer 3 Report

Comments and Suggestions for Authors

I appreciate the authors' efforts to revise their manuscript using my suggestions. I feel that it is now suitable for acceptance (although I did notice an extra apostrophe on line 101).